# Alexandria: Unsupervised High-Precision Knowledge Base Construction using a Probabilistic Program

**John Winn**                                               jwinn@microsoft.com
**John Guiver**                                         joguiver@microsoft.com
**Sam Webster**                                            sweb@microsoft.com
**Yordan Zaykov**                                       yordanz@microsoft.com
**Martin Kukla**                                         makukl@microsoft.com
**Dany Fabian**                                          danfab@microsoft.com
*Microsoft Research, 21 Station Road, Cambridge CB1 2FB, UK*

## Abstract

Creating a knowledge base that is accurate, up-to-date and complete remains a significant challenge despite substantial efforts in automated knowledge base construction. In this paper, we present Alexandria – a system for unsupervised, high-precision knowledge base construction. Alexandria uses a probabilistic program to define a process of converting knowledge base facts into unstructured text. Using probabilistic inference, we can invert this program and so retrieve facts, schemas and entities from web text. The use of a probabilistic program allows uncertainty in the text to be propagated through to the retrieved facts, which increases accuracy and helps merge facts from multiple sources. Because Alexandria does not require labelled training data, knowledge bases can be constructed with the minimum of manual input. We demonstrate this by constructing a high precision (typically 97%+) knowledge base for people from a single seed fact.

## 1. Introduction & related work

Search engines and conversational assistants require huge stores of knowledge in order to answer questions and understand basic facts about the world. As a result, there has been significant interest in creating such knowledge bases (KBs) and corresponding efforts to automate their construction and maintenance (see [Weikum and Theobald, 2010] for a review). For example, KnowledgeVault [Dong et al., 2014], NELL [Carlson et al., 2010, Mitchell et al., 2015], YAGO2 [Hoffart et al., 2013], DIG [P. Szekely et al., 2015], and many other systems aim either to construct a KB automatically or make an existing KB more complete. Despite these efforts, there remain significant ongoing challenges with keeping KBs up-to-date, accurate and complete. Existing automated approaches still require manual effort in the form of at least one of the following: **a provided set of entities** used for supervised training of components such as entity linkers/recognizers; **a provided schema** used to define properties/relations of entities; or **a provided set of annotated texts** used to train fact extractors/part of speech taggers. The holy grail of KB construction and maintenance would be a system which could learn and update its own schema, which could automatically discover new entities as they come into existence, and which could extract facts from natural text with such high precision that no human checking is needed.

With this goal in mind, we present Alexandria – a system for unsupervised, high-precision knowledge base construction. At the core of Alexandria is a probabilistic program that

defines a process of generating text from a knowledge base consisting of a large set of typed entities. By applying probabilistic inference to this program, we can reason in the inverse direction: going from text back to facts. This inverse reasoning allows us to retrieve facts, schemas and entities from web text. The use of a probabilistic program also provides an elegant way to handle the uncertainty inherent in natural text. An important advantage of using a generative model is that Alexandria does not require labelled data, which means it can be applied to new domains with little or no manual effort. The model is also inherently task-neutral – by varying which variables in the model are observed and which are inferred, the same model can be used for: learning a schema (relation discovery), entity discovery, entity linking, fact retrieval and other tasks, such as finding sources that support a particular fact. In this paper we demonstrate schema learning, fact retrieval, entity discovery and entity linking. We will evaluate the former two tasks, while the latter two are performed as part of these main tasks.

An attractive aspect of our approach is that the entire system is defined by one coherent probabilistic model. This removes the need to create and train many separate components such as tokenizers, named entity recognizers, part-of-speech taggers, fact extractors, linkers and so on; a disadvantage of having such multiple components is that they are likely to encode different underlying assumptions, reducing the accuracy of the combined system. Furthermore, the use of a single probabilistic program allows uncertainty to be propagated consistently throughout the system – from the raw web text right through to the extracted facts (and back).

**Related work** – There has been a significant amount of work on automated knowledge base construction [Weikum and Theobald, 2010]. Because the Alexandria system can be used to perform many different tasks, it is related to a range of previous, task-specific systems. Here we describe the most relevant.

*Unsupervised learning* – Open IE (Information Extraction) systems, such as Reverb [Fader et al., 2011] and OLLIE [Mausam et al., 2012] aim to discover information across multiple domains without having labelled data for new domains. Such systems do not have an underlying schema, but instead retain information in a lexical form. This can lead to duplication of the same fact stored using different words, or can even allow conflicting facts to be stored. Representing facts in lexical form makes them hard for applications to consume, since there is no schema to query against. In contrast, Alexandria aims to infer the underlying schema of new domains and to extract facts in a consistent form separate from their lexical representation.

*Schema learning* – the closest existing work is Biperpedia [Gupta et al., 2014] which aims to discover properties for many classes at once. Biperpedia uses search engine query logs as well as text to discover attributes, in a process that involves a number of trained classifiers and corresponding labelled training data. Alexandria's key differences are its unsupervised approach and the fact that schema learning is integrated into a single probabilistic model also used to perform other tasks.

*Web scale fact extraction* – several existing systems can extract facts against a known schema across the entire web. These include KnowledgeVault [Dong et al., 2014], NELL [Mitchell et al., 2015] and DeepDive [Zhang, 2015]. Of these, KnowledgeVault is the largest scale and has performed KB completion (filling in missing values for entities where most values are known) of over 250M facts. DeepDive is perhaps the most similar system to Alexandria in

```
// Loop over properties in the schema
for(int i=0;i<props.Length;i++) {
 // Pick # names from geometric dist
 int numNames=random Geometric(probNames);
 // Allocate array for storing names
 var names=new string[numNames];
 // Pick names from property name prior
 for(int j=0;j<numNames;j++) {
  names[j]=random Property.NamePrior;
 }
 // Set generated strings as property names
 props[i].Names=names;

 // Pick one built-in type prior at random
 var typePrior=random Uniform(typePriors);
 // Draw a type instance from the type prior.
 props[i].Type=random typePrior;
}
```
(a)

```
// Create set of entities (of the same type)
Entity[] entities = new Entity[entityCount];
// Loop over entities of this type
for(int j=0;j<entities.Length;j++) {
  // Loop over properties in the schema
  for(int i=0;i<props.Length;i++) {
    // Pick # alts from geometric dist
    int numAlts = random Geometric(probAlt);
    object[] alts = new object[numAlts];
    // Loop over alternatives
    for(int k=0;k<alts.Length;k++) {
      // Choose a property value from prior
      alts[k]=random props[i].Type.Prior;
    }
    // Set alternatives as property value
    entities[j][i]=alts;
  }
}
```
(b)

Figure 1: Programs for generating (a) a schema for an entity type (b) a KB of entities.

that it is based around a probabilistic model – a Markov Logic Network (MLN) [Richardson and Domingos, 2006]. DeepDive uses hand-constructed feature extractors to extract candidate facts from text for incorporation into the MLN. Because Alexandria uses a generative model of text, it can be applied directly to web text, without the need for feature extractors; in the mode described in this paper only a single seed fact is needed.

## 2. The Probabilistic Program

In recent years, there has been much interest in probabilistic programming as a powerful tool for defining rich probabilistic models [Defense Advanced Research Projects Agency, 2013, Daniel Roy et al., 2017]. The Alexandria probabilistic program makes full use of this power, through language features such as polymorphism, objects and collections. To our knowledge, this probabilistic program is also the largest and most complex deployed in any application today. Using a rich, complex probabilistic program allows the process of generating text from facts to be modelled very precisely, leading to high accuracy. The complete program generates: a schema for an entity type, which consists of a set of named, typed properties; a knowledge base, with a large set of typed entities; and a number of text extracts, each describing an entity from the knowledge base in natural language. We use an extended form of the Infer.NET inference engine [Minka et al., 2018] to invert this probabilistic program, and so infer a schema and knowledge base given a very large number of web text extracts. A major contribution of this work is to show that such large scale inference can be performed in complex probabilistic programs, as will be discussed in Section 4.

We show probabilistic programs in C#, which allows for compact representation of the model. In these programs, the keyword random takes a distribution and returns an uncertain value with that distribution. The Uniform function takes a collection of objects and returns uniform distribution over these objects. Given these definitions, we now describe each section of the probabilistic program in detail.

**Generating a schema for an entity type** – The program in Figure 1a generates a schema for an entity type consisting of a set of properties, each with multiple names and a type. Names are drawn from a hand-constructed prior over property names. Types are drawn from a mixture over all built-in type priors. Types can have parameters, for example, a `Set` type has a parameter for the element type of the set – the type prior for each type defines a joint prior over these parameters, as described in Section 3. The core idea is that entity types are compound types learned from data, constructed from built-in primitive types whose parameters are also learned from data. These built-in types are the main way that prior knowledge is made available to the model.

**Generating a probabilistic knowledge base** – The program in Figure 1b generates an Alexandria knowledge base, consisting of a number of typed entities. Each typed entity has values for each property of the type – for example, an entity of type 'person' will have a value for the 'DateOfBirth' property. To allow for disagreement over the value of a property, an entity can have many alternative property values – for example, multiple dates of birth for a person entity where there is disagreement about exactly when they were born. In the probabilistic program, first `entityCount` entities are created. For each entity and property, the number of alternative values is drawn from a (1-based) geometric distribution with parameter `probAlt`. Each alternative values is then drawn from the prior for that property type.

Importantly, the alternative values for a property represent different possible *conflicting* values, only one of which could actually be true. Such alternative values are very different to sets of values, where many values can be true for the same person (such a person who is a singer *and* a songwriter). For this reason, we model sets very differently to alternatives, using a specific set type (see Section 3.4).

**Generating text from KB values** – Figure 2a summarises the process of converting typed values in the knowledge base into unstructured text describing those values. First, an

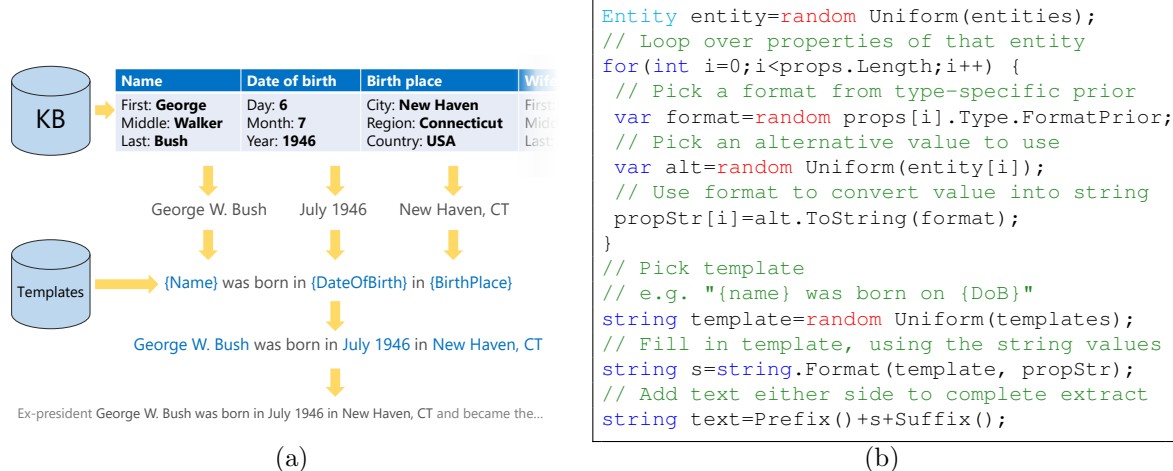

```
// Pick an entity to talk about at random
Entity entity=random Uniform(entities);
// Loop over properties of that entity
for(int i=0;i<props.Length;i++) {
 // Pick a format from type-specific prior
 var format=random props[i].Type.FormatPrior;
 // Pick an alternative value to use
 var alt=random Uniform(entity[i]);
 // Use format to convert value into string
 propStr[i]=alt.ToString(format);
}
// Pick template
// e.g. "{name} was born on {DoB}"
string template=random Uniform(templates);
// Fill in template, using the string values
string s=string.Format(template, propStr);
// Add text either side to complete extract
string text=Prefix()+s+Suffix();
```

(a)                                                          (b)

Figure 2: (a) Summary of the process of converting a typed entity from the KB into natural text containing facts about the entity. (b) Probabilistic program defining this process.

entity to describe is selected at random from the knowledge base. Then each of the entity's property values is converted into a string value using a format drawn from the type-specific format prior. For example, the date 6-July-1946 might get converted into the string "July 1946" using the format "MMMM yyyy". The next task is to embed these string property values into a natural sentence, or part of one. This embedding is achieved using a template, such as "{name} was born on {date_of_birth} in {place_of_birth}". The template is selected at random from a provided set of templates. These templates are drawn from a suitable prior over strings that enforces that braces occur at word boundaries. The template is filled in by replacing each property placeholder with the corresponding value string. Finally, suitable prefix and suffix strings are concatenated, allowing the completed template to appear inside a larger section of text. It is this final variable that will have observed values coming from the web and will allow the system to learn from web text. The probabilistic program defining this process is given in Figure 2b. Appendix C gives some illustrative examples from the many thousands of templates learned as part of the experiment for this paper.

The use of a probabilistic program to represent our knowledge base provides an elegant way to handle the uncertainty inherent in natural text. When we come to invert the probabilistic program and infer the values for a property, these values will be represented as probability distributions. As an example, if the text on the web page is "1972", then the extracted distribution will be uniform across all dates in 1972. Another example is if the text is "scientist" then the value will be a distribution over all professions which are scientists. Preserving this uncertainty is essential for maintaining very high precision and for correctly merging information from multiple sources.

## 2.1 Extensions to the core program

So far we have described the three sections of the core Alexandria probabilistic program. We have also made a number of extensions to this core program to improve the precision and recall of the system. These are illustrated in the probabilistic program given in Appendix A. The extensions are as follows:

**The property list model** – allows Alexandria to parse lists of property values rather than only values expressed in unstructured text. Each element in the list is assumed to refer to the name of the property as well as its value; a new kind of template, called a property list template, is used to represent such (name, value) pairs. Such templates allow property names to be learned during schema learning; see Appendix C for some examples.

**The page model** – allows Alexandria to associate together text extracts from the same HTML page or property list more strongly. Specifically it assumes that the number of entities referred to on a single page is much smaller than the total number of entities referred to in the entire web. To encode this assumption in a probabilistic program, a subset of all entities are first selected as entities on the page. All text extracts on the page can then only refer to these entities.

**The value noise model** – allows property values to vary slightly from one page to another while still being considered the same value. This model adds type-specific noise to a property value just before it is converted into text and written on the page. For the experiments in the paper, noise is applied only for `Quantity` properties, where multiplicative noise of $\pm 1\%$ is added. Modelling noise is particularly important for such numeric quantities,

like height, where the same person may have slightly different values reported on different pages. With this noise model in place, two pages reporting slightly different values are considered more likely to be referring to the same underlying entity, rather than less. It would also be possible to allow other kinds of noise for types other than `Quantity`. For example, for a `Date` type we could model noise such as mistakenly swapping the day and month in a date value.

## 3. The Built-in Property Types

The Alexandria probabilistic program defines an entity type as a set of properties, where each property has one of the built-in property types. The built-in types used in this paper are `Date`, `PersonName`, `Place`, `Hierarchy` and `Quantity`. There is also a set type `Set<T>` whose element type `T` can have any of the above types – for example, a set of people's names is written as `Set<PersonName>`. Finally, we have a 'catch-all' type that can generate any string value – this is used to discover properties which are not compatible with any of the built-in types (see Section 5).

These characteristics of each type are summarised in Table 1 and full details of each type are given in the following subsections. Each built-in type has a set of type parameters (such as the element type of a `Set`). A manually-specified `TypePrior` defines a distribution over instances of the type, including all type parameters. Every type defines a prior over values of the type (`Prior`), which may depend on the type parameters and so can be learned from data. In addition, each type has a `ToString(value,format)` method, implemented as a probabilistic program, which converts (uncertain) values of the type into strings. Finally, each type has a prior over format strings (`FormatPrior`) which defines a distribution over the format that can be passed in to the `ToString(value,format)` method.

The `Hierarchy` type takes a hierarchy as its type parameter. In this paper, we assume that a set of hierarchies have been manually provided and the type prior is a uniform distribution over these known hierarchies. For the experiments in Section 5, hierarchies were provided for occupations, nationalities, star signs, genders, religions, causes of death and hair & eye colors. In future we intend to learn these hierarchies by using a type prior over the structure and nodes of a hierarchy.

| Type | Type parameters | Prior | Example format string |
|------|-----------------|-------|----------------------|
| Date | - | Uniform over all dates between 1000 and 2100 | "{dd} {MMMM} {yyyy}" |
| PersonName | - | Uniform over valid names | "{First} {M}. {Last}" |
| Hierarchy | A hierarchy | Learned node probs | "Lower3" |
| Place | - | Learned place prob | "Lower3" |
| Quantity | Quantity kind (e.g. Length), Prior mean and variance | Learned Gaussian or log Gaussian | "{feet:F0}'{sub_inch:F0}" |
| Set<T> | Element type (T), Poisson, Beta parameters | Poisson over set size, Beta over mention prob. | "{0}, {1} and {2}" |
| 'catch-all' | - | Uniform over all strings | - |

Table 1: Summary of the built-in property types.

### 3.1 Object Types (Date, PersonName)

The `Date` and `PersonName` types are both *object types*. Object types have no type parameters and so there is only one instance of each object type. Object types have their own properties, for example the `Date` type has properties `Day`, `Month` and `Year`. For an object type, the value prior is the product of hand-specified priors over individual properties.

The `ToString()` method for object types requires that a set of format parts are manually specified. For `Date` these include: the date and month in numeric form with and without a leading zero (d,dd,M,MM); the short and long month names (MMM,MMMM), and the numeric year (yyyy). For people's names the format parts include first, middle and last names, nickname, initials, prefix and suffix. Given the format parts, the `ToString()` method for a object type is shown in Figure 6. This method uses `GetParts()` to compute string values for each format part from a value of the type – for example, computing the long and short month names from the month value. Each object type implements `GetParts()` using an appropriate probabilistic program. These are straightforward to write given appropriate probabilistic string operators [Yangel et al., 2017], so we have omitted them for compactness. The format prior is a uniform distribution over a manually specified set of valid formats, such as "dd MMMM yyyy".

### 3.2 Hierarchy Type

A `Hierarchy` type is used for properties that take one of a set of values. These values can be referred to with varying specificity through the use of a hierarchy, like the example in Figure 3. The leaf nodes of the hierarchy are the values of the type, such as "light blue", "yellow'. Nodes higher in the hierarchy allow for values to be referred to less precisely, such as "blue" vs "light blue". The `Hierarchy` type takes one type parameter which is the hierarchy itself. In this paper, we assume that a set of hierarchies have been manually provided and the type prior is a uniform distribution over these known hierarchies.

As shown in Figure 3, each node in the hierarchy has one or more strings associated with it. These strings are synonyms for that node value in natural language. A particular string can be associated with multiple nodes in a hierarchy, which allows ambiguous terms to be treated with appropriate uncertainty when the probabilistic program is inverted. The `ToString()` method for the `Hierarchy` type converts a value (leaf node) into a string according to a depth specified by the format as shown in Figure 7. Using this method, the

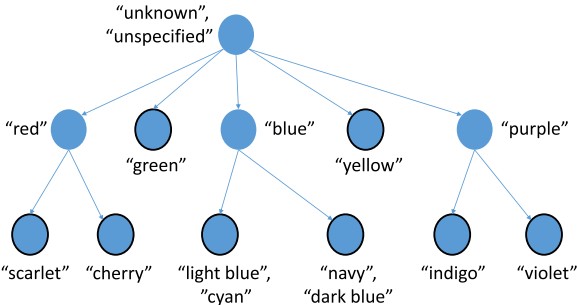

Figure 3: A toy example of a hierarchy used in a `HierarchyType`.

"cyan" leaf node can also appear in text as "light blue", "blue", "unknown" or "unspecified". Each leaf node also has a prior probability – these are included in the type parameters and so learned as part of schema learning.

The format prior for the `Hierarchy` type is uniform over formats of the form `{CaseFormat}{Depth}` where `CaseFormat` is one of *Default*, *Upper*, *Lower*, *Capitalized*, *FirstCapitalized*, and `Depth` is the depth in the hierarchy. The *Default* subformat uses the casing of the string value in the hierarchy, whereas the other case subformats convert to the specified case.

The `Place` type is a subtype of the `Hierarchy` type with a modified `ToString()` method which first selects one or more nodes along the path from the leaf node value to the root and then writes out each node in list form such as "Chelsea, London, England".

### 3.3 Quantity Type

A `Quantity` type is used for values that represents quantities such as lengths, weights and so on. The main type parameter determines which kind of quantity it is, out of the pre-defined set: {Length, Time, Weight}. Each kind of quantity has a set of units. Each unit has a conversion factor, a set of unit names (such as "m","metre","meter" etc.) and an optional sub-unit used for writing multi-unit quantities like "1m 58cm". The prior over values is Gaussian with type parameters for the mean and variance, along with a binary flag indicating if the prior is for the value or the logarithm of the value.

The process of converting `Quantity` values to strings handles both unit conversion and sub-units. This results in the more complex `ToString()` method of Figure 8. This method first extracts the unit and subunit and their individual numeric formats from the format string. The provided value is converted into the target unit using `InUnit()` and then into a string. If there is a sub-unit in the format, then the fractional part of the value is converted into the subunit using `NumberIn()` and again into a string. Finally, the value strings are inserted into the format string to give the returned result. The format prior for a `Quantity` type is uniform over a set of format strings derived from the quantity's unit and subunit names.

The sets of units used are manually provided to the system. However, we can learn different names for each unit from web text using a straightforward variant of the model where the unit names in the `ToString()` are replaced by random variables.

### 3.4 Set Type

The `Set<T>` type is used to represent sets of values of the above types. The primary type parameter is `T` which is the type of the elements in the set. The type prior over this element type is a uniform mixture of the type priors for all of the above types. An additional type parameter is the expected size of the set, used to define a Poisson distribution over the set size in the prior over values.

A value of a `Set<T>` type consists of a set of element values, each with a 'renown' probability representing how well-known the element value is for the entity. For example, a person may be well-known as a writer, but less well-known as a poet. A further two type parameters define the Beta distribution used as a prior over the renown probabilities. The `ToString()` method, shown in Figure 9, first creates a sample of elements from the

set, using each element's associated renown probability. These are the elements of the set that the author of the text knows about. The author then chooses (with equal probability) whether to mention all of these elements or just one element. The placeholder count in the format is then constrained to match the number of mentioned elements. Finally, the mentioned elements are converted into string values and inserted into the format string. The prior over formats for the `Set<T>` type allows up to ten elements connected by suitable separators such as comma, " and " and so on – for example "{0}, {1} and {2}"allows up to ten elements connected by suitable separators such as comma, " and " and so on – for example "{0}, {1} and {2}', though there is no limit on the size of the set value itself.

Explicitly modelling the cardinality of the set can be very helpful. For example, suppose two web pages both refer to Alan Smith but mention different pairs of parents. Knowing that people have two parents means that we can assume that these are two different Alan Smiths. Conversely, if the two pages mentioned different pairs of professions, say "actor, film director" and "comedian, screenwriter", it is still possible that they both refer to the same person, as people can have more than two professions.

## 4. Distributed Approximate Inference

Infer.NET allows probabilistic programs to be compiled into efficient C# code for inferring posterior distributions of specific variables in the program. This generated code applies a standard inference algorithm, such as expectation propagation (EP) [Minka, 2001], using a library of distributions and operations known as the Infer.NET runtime. For Alexandria, small examples of the core program can be compiled in this way and executed locally on a single machine. Inference over string values uses weighted automata, as described in [Yangel et al., 2017].

We need to run three kinds of large scale inference queries on the probabilistic program:
**Template learning** – where the set of `templates` is inferred, given web texts (`text`), the schema properties (`props`) and a set of `entities` where some of the property values are known;
**Schema learning** – where the set of schema properties (`props`) are inferred, given the web texts (`text`), a set of minimal `entities` (each with just a name and one other property) and a corresponding set of `templates`;
**Fact retrieval** – where the set of `entities` is inferred, given the entity names, the web texts (`text`), the schema properties (`props`) and a set of `templates`.

In essence, each of these queries infers one of the three variables (`props`, `templates`, `entities`) given the other two (and the web text). Fixing two out of these three is not essential, but helps to keep the inference process efficient and the schedule straightforward to parallelize.

We apply these queries to billions of documents and millions of entities. To achieve this scale, it was necessary to develop a distributed and optimised version of the inference algorithm. This distributed algorithm also uses the Infer.NET runtime but differs from the automatically generated algorithm in several important ways. Most importantly, it is written partially in SCOPE [Chaiken et al., 2008] allowing for large scale execution on Microsoft's Cosmos distributed computing platform. In addition, we use a manually defined, message-passing schedule optimised for rapid convergence.

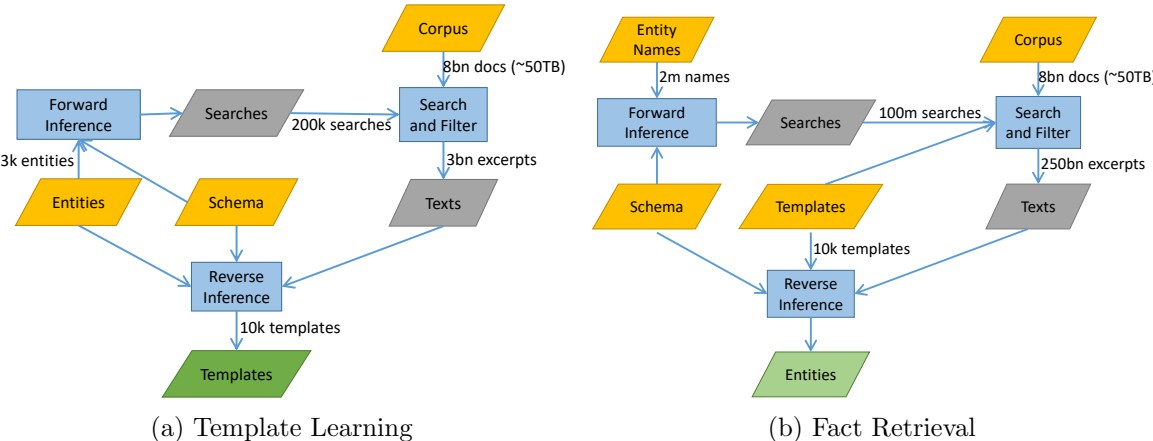

(a) Template Learning  (b) Fact Retrieval

Figure 4: Alexandria template learning and fact retrieval pipelines, with typical data sizes. Input data are shown in orange, interim data in grey, and output data in green.

Each built-in property type used a corresponding Infer.NET distribution type to represent uncertain values of the type and as messages in the message passing process. Values of these types can be highly structured (such as strings, objects, hierarchies and sets) and so distributions were chosen to be similarly structured. For example, distributions over entire objects and over entire sets were used. Distributions over hierarchical values were designed to exploit the structure of the hierarchy.

To gain significant speed ups, we applied additional approximations to the EP messages. These involved collapsing or removing uncertainty, where this could be done without adversely affecting precision. We found that it was essential to preserve uncertainty in which templates was matched (`template`), in the part of the matched text corresponding to a property (`propStr`) and in the extracted value (`alt`). Conversely, we found that we could collapse uncertainty in the `entity` being referred to in each piece of text, provided this was done conservatively and repeated at each iteration of inference. Each text was assigned to the most probable existing entity, provided the probability of this was above some high *mergeThreshold*, or otherwise to a new entity. Further speed ups were achieved by caching and re-using the results of certain slow message passing operations.

To cope with billions of documents, we first use forward inference in the program to compute an uncertain posterior over the `text` variable which defines all possible texts that could be generated by the program for a particular query. This posterior distribution is converted into a search which is executed at scale across the document corpus. The results of these searches are then returned as observations of the `text` variable. We apply these observations for reverse inference in the probabilistic program using the distributed inference algorithm. Figure 4 shows the processing pipelines for the Alexandria template learning and fact retrieval queries. The pipelines are annotated with typical data sizes for a large query.

These pipelines run on Microsoft's COSMOS distributed computing and Azure Batch cloud computing platforms. Execution times vary based on the task, the number of entities, and the number of properties. Note that template learning, which is more computationally intensive, can be done at a much lower frequency than fact retrieval.

.

## 5. Experiments and Results

We aim to demonstrate that Alexandria can infer both the schema for a type and high precision property values for entities of that type, from web scale text data. This inference is entirely unsupervised except for one single labelled example, used to bootstrap the system. The labelled example consists of the name "Barack Obama" and his date of birth "4 August 1961" which acts as the single textual label. Alexandria was also given a set of 3000 names used for schema learning and a separate test set of 4000 people's names to retrieve property values for. These test set names were selected at random from the top 1 million most common names searched for on Bing. Alexandria also has access to around 8 billion English HTML pages cached from the public web. To create observed values of the `text` variable, these pages are processed to remove HTML tags and non-visible text (giving about 50TB of text). Some tags, such as bullets or line breaks, are converted into suitable text characters, to allow the model to make use of the layout of the text on the page. In addition, sections of text which are exact duplicates are assumed to be copies and treated as a single observation, rather than multiple independent observations.

To bootstrap the system from a single example, we ran a small-scale query that infers both the `templates` and the schema properties (`props`), with `entities` observed to a set containing only our single known entity (Barack Obama) with property values for his name and date of birth. This bootstrapping process outputs a small set of 2-property templates and a corresponding schema. The process of inferring a full schema and set of property values then used the large scale inference queries described in Section 4, as follows:

1. Run **fact retrieval** to retrieve date of birth for 10 names from the schema learning dataset;
2. Run **template learning** given these 10 entities and the 2-property schema;
3. Repeat **fact retrieval/template learning** for 100 and then all 3000 names;
4. Run **schema learning** given the 3000 entities and templates learned from these;
5. Run **template learning** for the top 20 properties in the new schema using these 3000 entities;
6. Finally, run **fact retrieval** to retrieve values for the 4000 names in the separate test dataset.

These queries were run with `probAlt` set to 0.005, the *mergeThreshold* set to 0.99 and `entityCount` set to 10 million. For the 4000 person fact retrieval, retrieving relevant texts from 8bn documents took around 25K compute hours. Running distributed inference on these retrieved texts took a further 900 compute hours (about 2-3 hours of clock time). The largest fact retrieval that we have tried was for 2 million people. In this case, the text retrieval time increased sub-linearly to  250K compute hours, whilst the inference time increased almost linearly to around 200K hours (the reduction in time-per-entity is due to the reduced data for rarer entities).

**Results of schema learning** – Table 2 shows the top properties discovered during schema learning. The first column in Table 2 shows the most common inferred name for each property (other inferred names are shown in the final column). The second column shows the most probable inferred type, out of the built-in types from Section 3. The third column

| Inferred name | Inferred type | Entities | Domains | Other inferred names |
|---|---|---|---|---|
| name | PersonName | 2,964 | 4,545 | birth name, real name, birthname |
| born | Date | 2,756 | 3,471 | date of birth, birthday,... |
| birthplace | Place | 2,583 | 1,594 | place of birth, birth place, ... |
| occupation | Set<Hierarchy(Occupations)> | 2,569 | 801 | profession, occupations, ... |
| nationality | Set<Hierarchy(Nationalities)> | 2,485 | 505 | citizenship |
| zodiac sign | Hierarchy(StarSigns) | 2,336 | 328 | sign, star sign, zodiac sign, ... |
| gender | Hierarchy(Genders) | 2,110 | 247 | sex |
| spouse | Set<PersonName> | 2,058 | 665 | spouse(s), wife, husband, ... |
| hair color | Set<Hierarchy(HairColors)> | 2,050 | 340 | hair, natural hair colour |
| height | Quantity(Length) | 2,019 | 1,060 | |
| age | Quantity(Time) | 1,807 | 62 | |
| eye color | Set<Hierarchy(EyeColors)> | 1,705 | 405 | eyes, eye colour |
| parents | Set<PersonName> | 1,678 | 513 | father, mother, father name, ... |
| died | Date | 1,671 | 700 | date of death, death, death date |
| religion | Set<Hierarchy(Religions)> | 1,276 | 231 | |
| siblings | Set<PersonName> | 1,235 | 274 | brother, sister |
| children | Set<PersonName> | 1,121 | 368 | |
| weight | Quantity(Weight) | 594 | 325 | |
| cause of death | Hierarchy(CausesOfDeath) | 544 | 65 | |
| place of death | Place | 450 | 30 | location of death, death place |

Table 2: Results of schema learning for people. The top 20 discovered properties are shown, ordered by the number of entities with retrieved values for the property. See text for details of each column.

gives the number of entities where property values were found during schema learning. The rows of the table have been ordered by this count, which was also used to select the 20 properties shown. This ordering focuses on properties that most people have, rather than properties present only for certain kinds of people, such as musicians or sportspeople. The fourth column gives the number of web domains that referred to the property at least once. This is an indication of how generally relevant the property is. For this table, we have excluded niche properties which are referred to by fewer than 20 domains to focus on generally relevant properties.

Using the catch-all type, schema learning discovers some properties whose values are not compatible with any of the built-in types. These include descriptive text properties ('best known for', 'biography', 'quotations'), times and time ranges ('birth time', 'years active'), and a few rarer types. Mostly these would be straightforward to add as new built-in types.

**Evaluating fact retrieval** – Each retrieved value distribution needs to be evaluated against a ground truth value in the form of a string. Such evaluation is itself a challenging problem – for example, evaluating "scientist" against "physicist", "184cm" vs "6ft1½", "4/5/1980" against "May 4th 1980" and so on. To perform evaluation we re-use the probabilistic program and treat the ground truth text as an observed string value of the property. Inferring the property value gives a probability distribution of the same type as the retrieved value distributions. We then say that the value is correct if the expected probability of the ground truth value is higher under the retrieved value distribution than under the prior for that property type. So for a retrieved distribution $r$, a ground truth distribution $g$ and a prior $p$, the prediction is correct if $\sum_i g_i r_i > \sum_i g_i p_i$. where the sum is across all possible

values of the property. For example, if the ground truth is "1975" and the prediction is "5 May 1975" then the left hand side of the equation equals 1/365. Assuming a prior which is uniform over all dates between 1000 and 2500, the right hand side approximately equals 1/(365.25*1500). Thus, the LHS is greater than the RHS and the prediction is considered correct. If the prediction had been "5 May 1976", the LHS would equal zero and so the prediction would be considered incorrect. We aim to extract values at very high precision, around 97-99% or higher, and we evaluate our system against ground truth data from Microsoft's knowledge graph (Satori). Evaluating at such high precision raises some specific challenges:

- **Inherent uncertainty in the value** – for some entities there is uncertainty as to the value of certain properties. For example, about 1-2% of people have two different dates of birth each appearing in multiple web pages. In this situation a human judge is usually uncertain about which date is the correct one. If the ground truth data set can contain only one date, then a fact retrieval system may be unlucky and give the other one. A fairer approach is to allow the ground truth data set to contain both alternative values and consider either of these values to be correct. In our case, Satori already supports alternative values, and so we use these when evaluating our system.

- **Inherent quantitative uncertainty** – for numeric properties there is an inherent uncertainty arising from how accurately a value can be measured. For example, the height of a person has an inherent measurement uncertainty of about ±1cm. This uncertainty is accounted for in the noise model of the `Quantity` type (see Subsection 3.3). During evaluation we also use this noise model so that values are considered correct if they lie in a small neighbourhood of the ground truth value.

- **Set membership uncertainty** – for set properties, such as professions, some elements of the set are typically unambiguous (the best known profession, for example) but other elements are more subjective. For example, if a footballer has written an autobiography, does that also make her an author? For this reason we evaluate a set value as correct if any one predicted element is correct with respect to any ground truth element. To understand the accuracy of elements within each set, we also monitor element-wise precision and recall metrics.

For each name in the test set, Alexandria outputs a discovered set of entities with that name. For common names, this set can be quite large! We only wish to evaluate the one that corresponds to the entity in the test set, or none at all if that entity was not discovered. To do this, we again re-use the probabilistic program and treat the ground truth as a new observation, given the set of discovered entities. If the inferred most probable `entity` for the observation is one of the discovered entities, then that entity is used for evaluation. If the inferred `entity` is a new entity, then we record that no prediction was made for that ground truth entity, that is, it was not discovered by Alexandria.

**Results of fact retrieval** – Table 3 gives the evaluation metrics for fact retrieval for the discovered properties from Table 2. The fact retrieval process can result in more than one alternative value distribution for a particular name and property. For example, a date of birth property may have two retrieved alternatives, such as 5 May 1976 and 5 May 1977. Alternatives are ordered by the number of web pages that support them (which can be in the

hundreds or thousands). The metric 'precision@1' refers to the percentage of retrieved values where the first alternative was correct, that is, the alternative with the most supporting web pages. The metric 'precision@2' gives the percentage of retrieved values where the first or second alternatives were evaluated as correct. Recall is defined as the percentage of entities with ground truth values where a prediction was made. We also report the average number of alternatives ('Alts'), again where a prediction was made.

In considering Table 3, remember that these results were achieved without using *any* ground truth values, apart from a single date of birth value. Overall, the precisions of the first alternative (Prec@1) are high, with 13 of the 19 are in the 97%+ range and 9 of these above 98%. The lowest precisions are for children and hair and eye color properties, although these are still above 94%. Looking at the average number of alternatives, we see that most properties have just one alternative for the majority of predictions, so the precision@2 is the same as the precision@1. Exceptions to this include date of birth and date of death, which have 30-40% predictions with two alternatives (e.g. due to off-by-one errors). Considering the second alternative increases the precision of these properties by about 1%, bringing them to over 99% precision. The recall of the properties varies widely, from 16.4% for siblings through to 95.3% for date of birth, with an average of 59.7%. Some of this variation is due to the maximum possible recall varying, that is, the variation in the fraction of values actually available on the web. Another factor is how well a set of templates can captures how such values are expressed in text. For example, dates of birth and death are often expressed in a standard form, whereas there is more variation in how siblings and children are described.

| Property | Prec@1 | Prec@2 | Recall | Alts |
|---|---|---|---|---|
| born | 98.2% | 99.4% | 95.3% | 1.38 |
| birthplace | 96.6% | 97.4% | 76.4% | 1.09 |
| occupation | 97.1% | 97.3% | 79.4% | 1.19 |
| nationality | 98.2% | 98.2% | 83.3% | 1.01 |
| star sign | 96.6% | 97.3% | 28.3% | 1.08 |
| gender | 99.6% | 99.6% | 38.9% | 1.00 |
| height | 98.6% | 99.2% | 79.7% | 1.10 |
| hair color | 94.4% | 94.4% | 87.8% | 1.00 |
| spouse | 95.5% | 95.5% | 44.5% | 1.03 |
| age | 98.0% | 98.0% | 37.7% | 1.03 |
| died | 98.6% | 99.2% | 95.2% | 1.32 |

| Property | Prec@1 | Prec@2 | Recall | Alts |
|---|---|---|---|---|
| eye color | 94.7% | 94.7% | 88.4% | 1.00 |
| parents | 98.1% | 98.1% | 28.2% | 1.03 |
| religion | 97.6% | 97.6% | 57.5% | 1.00 |
| siblings | 100.0% | 100.0% | 16.4% | 1.01 |
| children | 94.3% | 94.3% | 17.2% | 1.05 |
| weight | 97.1% | 98.0% | 68.2% | 1.04 |
| cause of death | 98.4% | 98.4% | 63.4% | 1.04 |
| place of death | 97.9% | 97.9% | 47.7% | 1.00 |

Table 3: Fact retrieval metrics for the discovered properties.

**Comparison to previous results** – Most recent systems do not report results on any standard tasks, but instead use individual custom evaluation tasks and metrics, making rigorous comparison difficult. For example, YAGO2 [Hoffart et al., 2013] reports precisions in the range 92-98%, NELL [Mitchell et al., 2015] reports precisions in the range 80-85%, KnowledgeVault [Dong et al., 2014] reports a manually-assessed AUC of around 87%. The TAC-KBP Slot Filling track [Surdeanu and Ji, 2014] is a competition which aims to provide a comparison for fact retrieval systems – the highest precision for TAC2014 was 59% from the DeepDive system [Zhang, 2015]. Informal comparison to these reported results suggests that Alexandria's accuracy is at least as good as existing supervised systems and superior to that of existing unsupervised systems.

**Analysis of retrieval errors** – We manually checked the causes of a sample of errors. In decreasing order of frequency, the main causes were:

- A bad template match, where a normally reliable template retrieves the wrong value. This is sometimes caused by another property which often has the same value, such as 'residence' for 'place of birth'.

- Incorrect values on the web;

- Confusion between two people with the same name and several other properties also in common – often two members of the same family;

- Page semantics not preserved by the HTML processing; for example, two pieces of text which are well separated on the page end up adjacent in the processed text.

## 6. Conclusions & future work

In this paper, we have shown how Alexandria can perform schema learning and high-precision fact extraction unsupervised, except for a single seed example. Whilst the results in this paper are for people entities, the system has been designed to be generally applicable to many types of entity. It's worth noting that, in the process of learning about people, we have learned seed examples for other classes such as places, which we could use to do schema learning and fact extraction for these classes. By repeating this process recursively, we create the exciting possibility of using Alexandria like an Open IE system, to learn schemas, discovery entities and extract facts automatically across a large number of domains – this is our focus for future work. Our hope is that our high accuracy and strong typing will prevent 'drift' from occurring, which has reduced accuracy in previous Open IE systems.

The Alexandria model does not use any joint prior across property values – such as the graph prior and tensor factorization priors used in [Dong et al., 2014]. Incorporating such priors into the model has the potential to increase precision yet further.

Alexandria's template-based language model is relatively simple compared to some NLP systems used in related work. In contrast, Alexandria's model of types and values is in general more sophisticated, particularly in its handling and propagation of uncertainty. We believe that this has allowed the system to achieve very high precision. We expected to need a more sophisticated language model to achieve high recall – in fact, the ability to process the entire web means that we can still achieve good recall – a fact expressed in a complex way on one page is often expressed simply elsewhere.

The simplicity of the language model has one advantage – that it can be readily applied to text in many different languages. Indeed, we found that the system learned by itself to extract data from some non-English pages. To make full use of this would require making the built-in types multi-lingual, for example, allowing different month names in dates and different ways of writing numbers. The benefit would be to improve recall and also to learn how facts are expressed differently in different locales.

We believe that Alexandria makes a step towards the holy grail of completely automatic KB construction and maintenance – we look forward to trying out the system in many new domains to see if the successful unsupervised learning of people can be replicated for other entity types.

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

# Appendices

## A. Program for extensions to core model

Figure 5 gives the probabilistic program for the three extensions to the core model described in Subsection 2.1

```
// Pick a subset of entities for this page
var pageEnts=random Subset(entities,numOnPage);
// Generate a set of texts within the page
for(int j=0;j<texts.Length;j++) {
  // Pick an entity to talk about
  var entity=random Uniform(pageEnts);
  // Pick a property index
  int i=random Uniform(props.Length);
  // Pick a format from type-specific prior
  var format=random props[i].Type.FormatPrior;
  // Pick an alternative value to use
  var alt=random Uniform(entity[i]);
  // Add type-specific noise
  var noisyVal=props[i].Type.AddNoise(alt);
  // Use format to convert value into string
  var str=noisyVal.ToString(format);
  // Pick a property list template e.g. "|{propName}: {propValue}|"
  string t=random Uniform(listTemplates);
  // Fill in the property name and value
  texts[j]=string.Format(t,props[i].Name,str);
}
```

Figure 5: Probabilistic program demonstrating a number of extensions to the core model.

## B. Programs for Alexandria types

Figure 6 to Figure 9 show the probabilistic programs for the `ToString(value,format)` methods of the Alexandria types described in Section 3.

```
string ToString(object value, string format){
  // Get strings for each format part from the value objects
  string[] partStrs=GetParts(value);
  // Insert part strings into format string such as "{dd} {MMMM} {yyyy}"
  return string.Format(format, partStrs);
}
```

Figure 6: Implementation of `ToString()` for an object type.

```
string ToString(Node value, string format) {
  // Get depth e.g. "{Depth0}" gives 0.
  int depth=GetDepthFromFormat(format);
  // Get ancestor at that depth
  Node ancestor=value.GetAncestorAt(depth);
  // Pick one of the texts at this node
  return random Uniform(ancestor.Texts);
}
```

Figure 7: Implementation of `ToString()` for the `Hierarchy` type.

```
string ToString(Quantity value, string format)
{
  // Get unit from the format, such as "{m:F2}m" or "{feet:F0}'{sub_inch:F0}"
  Unit unit=GetUnit(format,out unitFormat);
  // Get subunit from format (if any)
  Unit sub =GetSubUnit(format,out subFormat);
  // Convert value into the target unit
  double d=value.InUnit(unit);
  // Format numeric value into a string
  string unitStr=d.ToString(unitFormat);
  if (sub==null) {
    return string.Format(format, unitStr);
  }
  // Sub-unit conversion
  double d2=Math.Frac(d)*sub.NumberIn(unit);
  // Value in sub-unit as string
  string subStr=d2.ToString(subFormat);
  return string.Format(format,unitStr,subStr);
}
```

Figure 8: Implementation of `ToString()` for the `Quantity` type.

```
string ToString(Set<Elem<T>> set,string format)
{
  // Sample elements to mention
  var vals=new List<T>();
  foreach(Elem<T> el in set) {
    bool mention = random Bernoulli(el.Prob);
    if (mention) vals.Add(el.Value);
  }
  vals.Permute(); // Permute order of values
  // With probability 0.5, keep only one value
  if (random Bernoulli(0.5)) vals=vals.Take(1);
  // Get placeholder count for format. e.g. "{0}, {1} and {2}" gives 3.
  int count=GetPlaceholderCount(format);
  // Constrain the format to match the sample
  constrain(count==vals.Count);
  // Convert values to strings & embed in format
  var strs=vals.Select(el=>el.ToString());
  return string.Format(format,strs);
}
```

Figure 9: Implementations of `ToString()` for the `Set` type.

## C. Example Templates

Table 4 gives some illustrative examples from the many thousands of templates learned as part of the experiment for this paper. Some of these templates contain special characters which represent document structure collapsed from the original HTML; the meaning of these structure characters is given in the structure key.

In addition, Table 4 includes some examples of nested templates, where one template is a substring of another. Such nested templates are treated as separate by the model which has the advantage that they can be assigned different template probabilities. Longer templates are generally less confusable with other similar templates, which leads to lower uncertainty in the `template` variable during inference.

| Text templates | Structure key: | ⏚ | Break |
|---|---|---|---|
| {age}-old {name} | | ❖ | Section |
| {name}'s sister {siblings} | | ● | Bullet |
| {name} was born in {place_of_birth}. Her mother, {parents} | | ▎ | Vertical separator |
| {name} stands {height} tall and weighs {weight} | | ‖ | Header separator |
| {name} died from {cause_of_death} in {date_of_death} | | ▬ | Horizontal separator |
| {occupations} {name} ({date_of_birth}
{occupations} {name} ({date_of_birth}-{date_of_death}
{nationalities} {occupations} {name} ({date_of_birth}-{date_of_death} | *These three templates are nested - shorter ones are substrings of longer ones.* | | |
| **Text templates with document structure** | | | |
| {name}▬Famous As:▎{occupations}▬Date of Birth:▎{date_of_birth}▬Place of Birth:▎{place_of_birth}▬Height:▎{height} | | | |
| {name}●Born❖{date_of_birth}●Birthplace❖{place_of_birth}●Spouse❖{spouses} | | | |
| **Property list templates** | | | |
| Born: {date_of_birth} Birthplace: {place_of_birth}. Gender: {gender} Religion: {religion} | | | |
| Died: {date_of_death}, in {place_of_death} | | | |
| Kids: {children} | | | |

Table 4: Example learned templates