# OpenReview forum: "Alexandria: Unsupervised High-Precision Knowledge Base Construction using a Probabilistic Program"
_AKBC.ws/2019/Conference — AKBC 2019_

### Official Review · AnonReviewer1 · 2018-12-16
**Very interesting overview of an existing knowledge base, will inspire interesting discussions**

**Rating:** 9
**Confidence:** 4

**Review:**


The paper is about a knowledge base that is constructed with a probabilistic model (Markov logic network, as implemented in Infer.NET). The system is expansive and covers many important aspects, such as data types and values, templates for extraction and  generation, noise models. The model addresses three kinds of large scale inferences: (1) template learning, (2) schema learning, and (3) fact retrieval.   The knowledge base construction approach is evaluated against other knowledge base approaches, YAGO2, NELL, Knowledge Vault, and DeepDive.

The paper is a very interesting overview over the knowledge base Alexandia, that would inspire interesting discussions at the AKBC conference.

My only pet peeve are a set of initial claims, intended to be distinguishing this approach from others, which are simply not true:
-  "Alexandia's key differences are its unsupervised approach" -- clearly Alexandria requires prior knowledge on types, and furthermore weakly supervision for template learning etc. Unsupervised means "not require any clue about what is to be predicted".
- "KnowledgeVault cannot discover new entities" -- I'd be surprised.
- "DeepDive uses hand-constructed feature extractors" -- It is a matter of preferences whether one seeds the extraction with patterns or data. This statement does not convince me.
While the discussion of these differences is important, I suggest the authors use more professional language.

---

> ### Author Response · Authors · 2019-01-22
> **Response to review comments for AnonReviewer1**
>
> Thank you for your helpful review. Here are some responses to specific questions:
>
> KnowledgeVault:
> Thank you for bringing this to our attention. We will modify our statement in the updated paper.
>
> DeepDive. "It is a matter of preferences whether one seeds the extraction with patterns or data.":
> This is true. The point we are trying to make here is that we are starting with a single well-known fact with no manual effort involved.
>
> "Alexandria's key differences are its unsupervised approach":
> "Unsupervised learning is a branch of machine learning that learns from test data that has not been labelled, classified or categorized" (Wikipedia). That is the case in the experiment reported in this paper except for the provision of a single data point. I guess the point you're making is that the knowledge inherent in automatically setting sensible priors on properties (for example heights of people) requires a set of values for the property. But in the paper experiment, such priors are learnt by type inference on extracted values as part of schema learning, and so, again, do not need labelled data. The types themselves are general and not domain or property or data specific. Hierarchies are hand-built in this experiment and therefore should be considered domain-specific prior information, but we have research addressing hierarchy learning which would, again, be fed by extracted values.

---

### Official Review · AnonReviewer3 · 2019-01-10
**Learning to extract facts with little prior except a structural prior from a probabilistic program**

**Rating:** 8
**Confidence:** 4

**Review:**

This paper uses a probabilistic program describing the process by which facts describing entities can be realised in text, and a large number of web pages, to learn to perform fact extraction about people using a single seed fact.

Despite the prodigious computational cost (close to a half million hours of computation to acquire a KB only about people) I found the scale at which this paper applied probalistic programming exciting.  It suggests that providing a structural prior in this form, and then learning the parameters of a model with that structure is a practical technique that could be applied widely.

Questions that arose whilst reading:  the most direct comparison was with a system using Markov Logic Networks, in which the structural prior takes the form of a FOL theory. A more direct comparison would have been useful - in particular, an estimate of the difficulty of expressing and equivalently powerful model, and the computational cost of trainining that model, in MLN.

Quite a lot of tuning was required to make training tractable (for outlying values of tractable) - this limits the likely applicability of the technique.

The paper suggests in future work an extension to open domain fact extraction, but it is not clear how complex or tractable the require prob program would be. The one in the paper is in some respects (types mainly) specific to the facts-about-people setting.

It is unclear why theTAC-KBP Slot Filling track  was mentioned, given that performance on this track does not seem to have been evaluated. An informal evaluation suggesting beyond SoA performance is mentioned, but not useful details given. This significantly weakens what otherwise could be a stand-out paper

---

> ### Author Response · Authors · 2019-01-22
> **Response to review comments for AnonReviewer3**
>
> Thank you for your helpful review. Here are some responses to your specific questions:
> 1. Comparison with MLN: Probabilistic programs are substantially richer than MLNs - and indeed can express MLNs as a special case. It is unclear if an MLN can be used to express our model - we suspect not, since our model can, for example, reason about the cardinality of sets of values which cannot be achieved in a compact form with FOI. Expressing an equivalently powerful model would be a substantial research exercise in itself!
> 2. "Quite a lot of tuning was required": Could you clarify what you mean by tuning? Are you referring to the approximate inference assumptions? If so, this is 'done'. There is no need to do this for each domain. If you are referring to computational performance, the computation is expensive due to running on the whole web, and dramatic speed-ups can be obtained by restricting to particular web domains.
> 3. Extensibility to other domains: It is true that the hierarchies in the paper are people-specific, but we are developing the ability to learn these. Otherwise, the types in the paper have broad applicability, and they (along with a few others which have been added more recently) have since been used for 100's of entity types.
> 4.  "It is unclear why the TAC-KBP Slot Filling track was mentioned": We wanted to evaluate our system against the TAC-KBP Slot Filling track but were unable to do so since the track required manual evaluation.  Whilst we could have trained comparable judges to judge our system there is significant subjective judgement needed and using different judges would bring any comparison into question.  What is needed is automatic evaluation of such systems to allow ongoing fair comparison.

---

### Official Review · AnonReviewer2 · 2019-01-10
**Although the paper has a lot of promise, it is not suitable for publication in an academic conference in its present form, in my view. I advise the authors to review all the comments, and prepare a significantly revised version for future publication.**

**Rating:** 6
**Confidence:** 3

**Review:**


The authors present Alexandria, a system for unsupervised, high-precision knowledge base construction. Alexandria uses a probabilistic program to define a process of converting knowledge base facts into unstructured text. The authors evaluate Alexandria by constructing a high precision (typically 97%+) knowledge base for people from a single seed fact.

Strengths:

--although it is unusual to combine and intro and related work into a single section, I enjoyed the authors' succinct statement (which I agree with) about the 'holy grail of KB construction'. Overall, I think the writing of the paper started off on a strong note.

--the paper has a lot of promise in considering a unified view of KB construction. Although I am not recommending an accept, I hope the authors will continue this work and submit a revised version for future consideration (whether in the next iteration of this conference, or another)

Weaknesses:

--It is not necessary to be putting 'actual programs' (Figure 1). It does not serve much of a purpose and is inappropriate; the authors should either use pseudocode, or just describe it in the text.

--In Web-scale fact extraction, the domain-specific insight graphs (DIG) system should also be mentioned and cited, since it is an end-to-end system that has been used for KGC in unusual domains like human trafficking.

--Starting from Section 2, the whole paper starts reading a bit like a technical manual/specification document. This kind of paper is not appropriate for an academic audience; the paper should have been written in a much more conceptual way, with actual implementations/programs/code details relegated to a github repo/wiki, and with a link to the same in the paper.

---

> ### Author Response · Authors · 2019-01-22
> **Response to review comments for AnonReviewer2**
>
> Thank you for your helpful review. Here are some responses to your specific questions:
> 1. "It is not necessary to be putting 'actual programs' (Figure 1). It does not serve much of a purpose and is inappropriate; the authors should either use pseudocode, or just describe it in the text.": We are using a probabilistic programming approach to construct the knowledge base.  The programs here are not source code for implementation but define the probabilistic models used - in others words the core assumptions of our method.  It is common practice to share programs in a probabilistic programming paper, for example: Picture: A Probabilistic Programming Language for Scene Perception (Kohli et al., 2015) or Human-level concept learning through probabilistic program induction (Lake et al., 2015) both show example programs.
> 2. "Mention and cite DIG": Thank you for bringing this to our attention. We will certainly cite this in an updated version. DIG has several aspects. (1) The training of domain-specific feature extractors from semi-structured pages using user-specified examples and automatically generated examples for crowd-sourcing, (2) The mapping of knowledge into the DIG ontology including the preservation of provenance , (3) Entity linking using a variety of similarity algorithms on text and images, and (4) Querying and Visualization. In Alexandria both (1) and (3) are solved by running inference in the same probabilistic model, and both unstructured data and semi-structured source are supported. But image data is not currently supported. (2) and (4) are not directly addressed by the Alexandria system.
> 3. "The whole paper starts reading a bit like a technical manual/specification document.":  As stated above, these programs are not implementations but probabilistic programs defining the models/assumptions used to learn the knowledge base.  They are a key contribution of this paper which is why we have included them to allow academic readers to assess and critique assumptions encoded in the system.

---

### Meta-Review · Area_Chair1 · 2019-02-12
**A novel approach to KG construction that shows initial promising results**

**Recommendation:** Accept (Oral)
**Confidence:** 4

**Metareview:**

The authors propose Alexandria, a probabilistic programming approach to AKBC. The core idea of the approach is to use a generative probabilistic program to model natural language expressions of facts using templates, and then reverse this process to learn new templates and properties from text. Evaluation for a small domain showed promising results.

The critical consensus was that this paper presents an interesting idea and should be accepted. There were several concerns about the representation of related work and some concerns about the evaluation and results, particularly the restriction to a single domain and the computational costs of the system, and criticism about the balance of high-level and low-level ideas in the writing. The authors have addressed some of these concerns in the rebuttal and in a draft revision.

---

### Decision · Program_Chairs · 2019-02-15
**AKBC 2019 Conference Decision**

Accept